# Effects of anti-SSA antibodies on the response to methotrexate in rheumatoid arthritis: A retrospective multicenter observational study

Daisuke Waki[1]*, Hiroya Tamai[2], Ritsuko Yokochi[3], Toshiki Kido[4], Yuriko Yagyu[5], Ryo Yanai[6], Ken-Ei Sada[7]

1 Department of Endocrinology and Rheumatology, Kurashiki Central Hospital, Okayama, Japan, 2 Division of Rheumatology, Department of Internal Medicine, Keio University School of Medicine, Tokyo, Japan, 3 Division of Hematology and Rheumatology, Teikyo University Chiba Medical Center, Chiba, Japan, 4 The First Department of Internal Medicine, Toyama University Hospital, Toyama, Japan, 5 Department of Internal Medicine, Tokyo Kyōsai Hospital, Tokyo, Japan, 6 Division of Rheumatology, Showa University Hospital, Tokyo, Japan, 7 Department of Clinical Epidemiology, Kochi Medical School, Kochi University, Kochi, Japan

◉ These authors contributed equally to this work.
* watin820@gmail.com

## Abstract

Comparison of clinical response to methotrexate between anti-SSA antibody-positive and -negative patients with methotrexate-naïve rheumatoid arthritis and investigate the reasons for the differences in the response. For this multicenter retrospective cohort study, a total of 210 consecutive patients with rheumatoid arthritis who newly initiated methotrexate were recruited. The effects of anti-SSA antibody positivity on achieving a low disease activity according to the 28-joint Disease Activity Score based on C-reactive protein after 6 months of methotrexate administration were investigated using a logistic regression analysis. This study involved 32 and 178 anti-SSA antibody-positive and -negative patients, respectively. The rate of achieving low disease activity according to the 28-joint Disease Activity Score based on C-reactive protein at 6 months was significantly lower in the anti-SSA antibody-positive group than in the anti-SSA antibody-negative group (56.2% vs. 75.8%, $P = 0.030$). After 6 months, anti-SSA antibody-positive patients had significantly higher scores on the visual analogue scale (median [interquartile range]: 22 [15–41] vs. 19 [5–30], $P = 0.038$) and were frequently prescribed nonsteroidal anti-inflammatory drugs (37.5% vs. 18.0%, $P = 0.018$). In conclusion, the presence of anti-SSA antibodies might be a predictive factor for insufficient responses to treat-to-target strategy in rheumatoid arthritis. Residual pain might contribute to the reduced clinical response to methotrexate in anti-SSA antibody-positive patients with rheumatoid arthritis.

## Introduction

In the era of biologics, methotrexate (MTX) is still regarded as an anchor drug for the management of rheumatoid arthritis (RA). The treat-to-target (T2T) approach, including the early

**Data Availability Statement:** Data cannot be shared publicly because the data contain potentially identifying or sensitive patient information. However, data can be available from the Ethics

Committee of each hospital (contact via Ethics Committee of Keio University School of Medicine: med-rinri-jimu@adst.keio.ac.jp) for researchers who meet the criteria for access to confidential data.

**Funding:** The authors received no specific funding for this work.

**Competing interests:** The authors have declared that no competing interests exist.

induction of MTX, has contributed to an improvement in the rate of remission and low disease activity (LDA) [1, 2]. However, some patients still fail to achieve LDA despite the T2T approach [3, 4]. Several risk factors have been proposed as poor prognostic factors for the control of disease activity, including the presence of anti-cyclic citrullinated protein (anti-CCP) antibodies, rheumatoid factor (RF), and bone structural damage [5–9]. Several observational studies have demonstrated that anti-SSA antibody status can be a prognostic factor for a poor response to treatment, including tumor necrosis factor inhibitors, although other studies have shown conflicting results [10–12]. The discrepancies in the results might be because these studies did not consider the abovementioned poor prognostic factors or did not involve many patients with prolonged disease duration whose clinical presentation may have been altered by previous treatment(s). Furthermore, to the best of our knowledge, there has been no study on the response to MTX in MTX-naïve RA patients with or without anti-SSA antibodies.

Here, we conducted a multicenter observational study to analyze the differences in the clinical response of MTX-naïve RA patients in response to MTX, including anti-SSA antibody status and other poor prognostic factors.

## Materials and methods

### Patients

In this retrospective, multicenter, observational study, data were collected from the clinical records of adult RA patients newly initiated with MTX at four tertiary referral or university hospitals (Kurashiki Central Hospital, Teikyo University Chiba Medical Center, Keio University Hospital, and Toyama University Hospital). All patients fulfilled the 2010 diagnostic criteria of the American College of Rheumatology/European League Against Rheumatism (ACR/EULAR) or the 1987 diagnostic criteria of the ACR. The enrolled patients had never been treated with MTX or biologic disease-modifying antirheumatic drugs (bDMARDs) [13, 14]. Sjögren's syndrome (SS) was diagnosed according to the 2016 ACR/EULAR classification criteria [15].

We excluded patients according to the following criteria: patients who had not been tested for anti-SSA antibodies; patients who had not been assessed for disease activity at baseline or the following 6 months; patients who had been lost to follow-up; patients who failed to continue MTX for 6 months after starting MTX; patients who received more than 30 mg/day prednisolone equivalent corticosteroids within 6 months of MTX initiation; patients with fibromyalgia, connective tissue disease, or rheumatic musculoskeletal disease other than SS; and patients taking antidepressants, antipsychotics, or antidementia medication. Patients with extra-articular complications were also excluded from this study. Finally, we recruited 210 consecutive adult RA patients who newly initiated MTX.

Our study was performed according to the principles outlined in the Declaration of Helsinki and approved by the ethics committee of Keio University School of Medicine (approval number: 20200101). This study was also approved by the individual institutional review board of all participating hospitals. Informed consent from the patients was obtained by oral agreement or through opt-out in accordance with the regulations in Japan. The ethics committee in each hospital approved this opt-out consent mechanism.

### Antibody measurements

Anti-SSA antibody level was measured using the enzyme-linked immunosorbent assay (ELISA), chemiluminescent enzyme immunoassay (CLEIA), or fluorescence enzyme immunoassay (FEIA), with commercial assays from Medical & Biological Laboratories (Tokyo, Japan) or BML Inc. (Tokyo, Japan). The cut-off value was set at 7.0 U/ml for FEIA and 10.0 U/

ml for ELISA and CLEIA. Anti-CCP antibody was determined using a second-generation ELISA, and the cut-off level for positivity was set at 4.5 U/ml. IgM rheumatoid factor (IgM-RF) level was detected using a latex agglutination assay, and the cut-off level for positivity was set at 15 IU/ml.

### Data collection and definitions

Baseline data were collected within 2 weeks of the first MTX administration. Patient data were recorded at each follow-up visit and collected from medical records until 6 months after the first MTX administration. ΔTSJ was defined as the numeric difference between the numbers of tender joints and swollen joints. ΔPEG was defined as the numeric difference between the patient's visual analogue scale (VAS) score (patient VAS, 0–100 mm) and the evaluator's VAS score (physician VAS, 0–100 mm) [16]. Swollen and tender joints were counted according to the 28-joint Disease Activity Score (DAS28) [17]. We evaluated the disease activity using the DAS28-CRP, Simplified Disease Activity Index (SDAI), and Clinical Disease Activity Index (CDAI) [18]. For the DAS28-CRP, the cut-off point for remission and LDA was defined as <2.3 and ≤2.7, respectively [19]. The cut-off point for remission and LDA was defined as an SDAI of ≤3.3 and ≤11, respectively, and a CDAI of ≤2.8 and ≤10, respectively, according to the 2011 ACR/EULAR recommendation [20].

### Statistical analysis

The backgrounds of the anti-SSA antibody-positive and -negative groups were compared before multiple imputations. Differences between the groups were analyzed using Mann–Whitney U test or Student's *t*-test for continuous variables and Fisher's exact test for categorical variables. After multiple imputations for missing data, we performed a multivariable analysis using the data. After confirming that the distribution of missing values is not inconsistent with the assumptions of missing at random, 10 imputed datasets were generated. Logistic regression analysis for the rate of achieving LDA according to the DAS28-CRP at 6 months was adopted for all imputed datasets. The imputation procedure included all covariates included in this study. The results obtained for each dataset were pooled using Rubin's rules [21, 22]. The following variables were assessed as potential poor prognostic factors for the rate of achieving LDA based on the DAS28-CRP: age at RA onset, sex, anti-SSA antibody status, anti-CCP antibody status, and RF positivity. A sensitivity analysis for the rate of achieving LDA based on the DAS28-CRP at 6 months was also performed considering the significant difference in background factors between the anti-SSA antibody-positive and -negative groups as a factor. To explore other potential prognostic factors, we compared baseline characteristics between the two groups who achieved LDA based on the DAS28-CRP at 6 months and those that did not. We performed another sensitivity analysis considering those potential prognostic factors.

A difference was considered significant when the two-tailed *P*-value was <0.05. All analyses were performed using R statistical software (version 3.1.1, R Foundation for Statistical Computing, Vienna, Austria).

## Results

### Demographics

The baseline characteristics of the patients and their medications are presented in Table 1. Among the 210 study participants, 32 were anti-SSA antibody-positive. The anti-SSA antibody-positive and -negative groups were significantly different in terms of the proportion of women (87.5% vs. 68.5%, *P* = 0.033), IgM-RF positivity (75.0% vs. 54.0%, *P* = 0.032), anti-CCP

**Table 1. Patient characteristics at baseline.**

| | Missing n (%) | Anti-SSA antibody-negative group (n = 178) | Anti-SSA antibody-positive group (n = 32) | P |
|---|---|---|---|---|
| Women, n (%) | 0 (0) | 122 (68.5) | 28 (87.5) | 0.033 |
| Age at disease onset, year, median [IQR] | 0 (0) | 61.0 [52.0–72.0] | 58.0 [51.5–65.0] | 0.114 |
| Age at diagnosis, year, median [IQR] | 5 (2.4) | 63.0 [53.0–72.0] | 59.5 [52.8–67.3] | 0.170 |
| Disease duration, months, median [IQR] | 5 (2.4) | 5.0 [2.0–12.0] | 5.0 [2.0–13.0] | 0.909 |
| History of smoking, n (%) | 24 (11.4) | 65 (39.8) | 8 (32.0) | 0.515 |
| IgM-RF positivity, n (%) | 2 (1.0) | 95 (54.0) | 24 (75.0) | 0.032 |
| Anti-CCP antibody positivity, n (%) | 0 (0) | 101 (57.1) | 27 (84.4) | 0.003 |
| Sicca symptoms, n (%) | 51 (24.3) | 20 (14.8) | 8 (33.3) | 0.040 |
| Diagnosis of Sjögren's syndrome | 0 (0) | 0 (0) | 2 (6.2) | 0.023 |
| Steinblocker, n (%) | 2 (1.0) | | | 0.135 |
| I | | 133 (75.6) | 19 (59.4) | |
| II | | 29 (16.5) | 8 (25.0) | |
| III | | 2 (1.1) | 1 (3.1) | |
| IV | | 11 (6.2) | 3 (9.4) | |
| Patient VAS score, median [IQR] | 0 (0) | 40.0 [20.0–60.8] | 40.0 [27.3–61.3] | 0.519 |
| Physician VAS score, median [IQR] | 0 (0) | 25.0 [13.0–46.0] | 25.0 [20.0–35.0] | 0.770 |
| Number of tender joints, median [IQR] | 0 (0) | 2.0 [0–5.0] | 2.0 [0–4.0] | 0.849 |
| Number of swollen joints, median [IQR] | 0 (0) | 3.0 [1.0–5.0] | 3.5 [2.0–5.0] | 0.732 |
| ΔPEG, median [IQR] | 0 (0) | 7.0 [0–25.0] | 12.0 [0–28.0] | 0.396 |
| ΔTSJ, median [IQR] | 0 (0) | 0 [–3.0 to 1.0] | –1.0 [–2.0 to 0] | 0.447 |
| CRP, mg/dl, median [IQR] | 0 (0) | 0.54 [0.18–1.87] | 0.57 [0.12–1.46] | 0.460 |
| DAS28-CRP, mean ± SD | 0 (0) | 3.54 ± 1.23 | 3.52 ± 1.16 | 0.916 |
| CDAI, mean ± SD | 0 (0) | 14.53 ± 9.68 | 14.60 ± 9.12 | 0.971 |
| SDAI, mean ± SD | 0 (0) | 15.94 ± 10.63 | 15.80 ± 9.83 | 0.947 |
| Corticosteroid use, n (%) | 0 (0) | 43 (24.2) | 4 (12.5) | 0.172 |
| Corticosteroid dose, mg/day, mean ± SD | 0 (0) | 1.6 ± 3.4 | 0.59 ± 1.6 | 0.094 |
| NSAID use, n (%) | 0 (0) | 78 (43.8) | 7 (21.9) | 0.020 |
| Initial MTX dose, mg/week, mean ± SD | 0 (0) | 7.2 ± 1.2 | 7.5 ± 1.0 | 0.149 |
| csDMARD use, n (%) | 0 (0) | 52 (29.2) | 7 (21.9) | 0.522 |

Data are presented as median [interquartile range], mean ± standard deviation (SD), or n (%). IgM-RF, IgM rheumatoid factor; anti-CCP, anti-cyclic citrullinated protein; VAS, visual analogue scale; ΔPEG, the numeric difference between patient VAS score and physician VAS score; ΔTSJ, the numeric difference between numbers of tender and swollen joints; CDAI, Clinical Disease Activity Index; SDAI, Simplified Clinical Disease Activity Index; MTX, methotrexate; csDMARD, conventional synthetic disease-modifying antirheumatic drug; CRP, C-reactive protein; DAS28, 28-joint Disease Activity Score; NSAID, nonsteroidal anti-inflammatory drug.

antibody positivity (84.4% vs. 57.1%, $P = 0.003$), and sicca symptoms (33.3% vs. 14.8%, $P = 0.040$). Definite SS was only diagnosed in two patients because almost all patients with sicca symptoms never underwent a lip biopsy or ophthalmologic examination. The proportion of patients receiving nonsteroidal anti-inflammatory drugs (NSAIDs) at baseline was significantly lower in the anti-SSA antibody-positive group than in the negative group (21.9% vs. 43.8%, $P = 0.020$). Disease activity assessed by DAS28-CRP, CDAI, SDAI, and the initial MTX dose was not significantly different between the groups.

## Disease activity at 6 months after MTX administration

After 6 months, the proportion of patients who achieved LDA according to the DAS28-CRP was significantly lower in the anti-SSA antibody-positive group than in the corresponding antibody-negative group (56.2% vs. 75.8%, $P = 0.03$) (Table 2). In contrast, there was no

**Table 2. Disease activity and medications after 6 months of MTX administration.**

| | Missing n (%) | Anti-SSA antibody-negative group (n = 178) | Anti-SSA antibody-positive group (n = 32) | P |
|---|---|---|---|---|
| Patient VAS score, median [IQR] | 0 (0) | 19.0 [5.0–30.0] | 22.0 [15.0–41.3] | 0.038 |
| Physician VAS score, median [IQR] | 0 (0) | 10.0 [3.0–18.0] | 10.0 [3.0–18.5] | 0.698 |
| Number of tender joints, median [IQR] | 0 (0) | 0 [0, 1.0] | 0 [0, 2.0] | 0.475 |
| Number of swollen joints, median [IQR] | 0 (0) | 1.0 [0–2.0] | 1.0 [0–3.0] | 0.277 |
| ΔPEG, median [IQR] | 0 (0) | 5.0 [0–15.0] | 10.0 [2.8–25.5] | 0.053 |
| ΔTSJ, median [IQR] | 0 (0) | 0 [–1.0 to 0] | 0 [–2.3 to 1.0] | 0.944 |
| CRP, mg/dl, median [IQR] | 0 (0) | 0.12 [0.04–0.31] | 0.25 [0.05–0.50] | 0.150 |
| DAS28-CRP, mean ± SD | 0 (0) | 2.20 ± 0.86 | 2.48 ± 1.05 | 0.110 |
| CDAI, mean ± SD | 0 (0) | 5.78 ± 5.52 | 7.07 ± 6.37 | 0.237 |
| SDAI, mean ± SD | 0 (0) | 6.12 ± 5.79 | 7.50 ± 6.83 | 0.228 |
| Corticosteroid use, n (%) | 0 (0) | 43 (24.2) | 6 (18.8) | 0.651 |
| Corticosteroid dose, mg/day, mean ± SD | 0 (0) | 0.67 ± 1.50 | 0.67 ± 1.65 | 0.994 |
| NSAID use, n (%) | 0 (0) | 32 (18.0) | 12 (37.5) | 0.018 |
| MTX dose, mg/week, mean ± SD | 0 (0) | 10.8 ± 3.1 | 9.8 ± 3.9 | 0.048 |
| csDMARD use, n (%) | 0 (0) | 55 (30.9) | 11 (34.4) | 0.684 |
| DAS28-CRP remission, n (%) | 0 (0) | 114 (64.0) | 16 (50.0) | 0.166 |
| CDAI remission, n (%) | 0 (0) | 63 (35.6) | 9 (28.1) | 0.545 |
| SDAI remission, n (%) | 0 (0) | 67 (37.9) | 10 (31.2) | 0.553 |
| DAS28-CRP LDA, n (%) | 0 (0) | 135 (75.8) | 18 (56.2) | 0.03 |
| CDAI LDA, n (%) | 0 (0) | 145 (81.9) | 25 (78.1) | 0.624 |
| SDAI LDA, n (%) | 0 (0) | 144 (81.4) | 26 (81.2) | 1.000 |

Data are presented as median [interquartile range], mean ± standard deviation (SD), or n (%). VAS, visual analogue scale; ΔPEG, the numeric difference between patient VAS score and physician VAS score; ΔTSJ, the numeric difference between the numbers of tender and swollen joints; CDAI, Clinical Disease Activity Index; SDAI, Simplified Clinical Disease Activity Index; MTX, methotrexate; csDMARD, conventional synthetic disease-modifying antirheumatic drug; LDA, low disease activity; CRP, C-reactive protein; DAS28, 28-joint Disease Activity Score; NSAID, nonsteroidal anti-inflammatory drug.

significant difference between the groups in the proportion of remission based on the DAS28-CRP, SDAI, and CDAI, and in the proportion of patients with LDA according to the SDAI and CDAI. The patient VAS score (median [interquartile range]: 22.0 [15.0–41.3] vs. 19.0 [5.0–41.3], P = 0.038) and the prevalence of NSAID use (37.5% vs. 18.0%, P = 0.018) were significantly higher in the anti-SSA antibody-positive group than in the anti-SSA antibody-negative group. ΔPEG tended to be higher in the anti-SSA antibody-positive group than in the corresponding antibody-negative group (median [interquartile range]: 10.0 [2.8–25.5] vs. 5.0 [0–15.0], respectively, P = 0.053) (Fig 1). The change in patient VAS scores before and 6 months after MTX administration for each patient is shown in Fig 2.

## Multivariable analysis for achieving LDA based on the DAS28-CRP

After multiple imputations for missing values, the logistic regression analysis showed that anti-SSA antibody positivity was significantly associated with failure to achieve LDA according to the DAS28-CRP at 6 months, even after adjusting for the potential poor prognostic factors IgM-RF, anti-CCP antibody, age, and sex (odds ratio: 0.431, 95% confidence interval: 0.190–0.978, P = 0.044) (Table 3).

## Sensitivity analysis for achieving LDA based on the DAS28-CRP

Logistic regression analysis was performed after multiple imputations for missing values considering sex, MTX dose at 6 months, IgM-RF, and anti-CCP antibody as confounding factors,

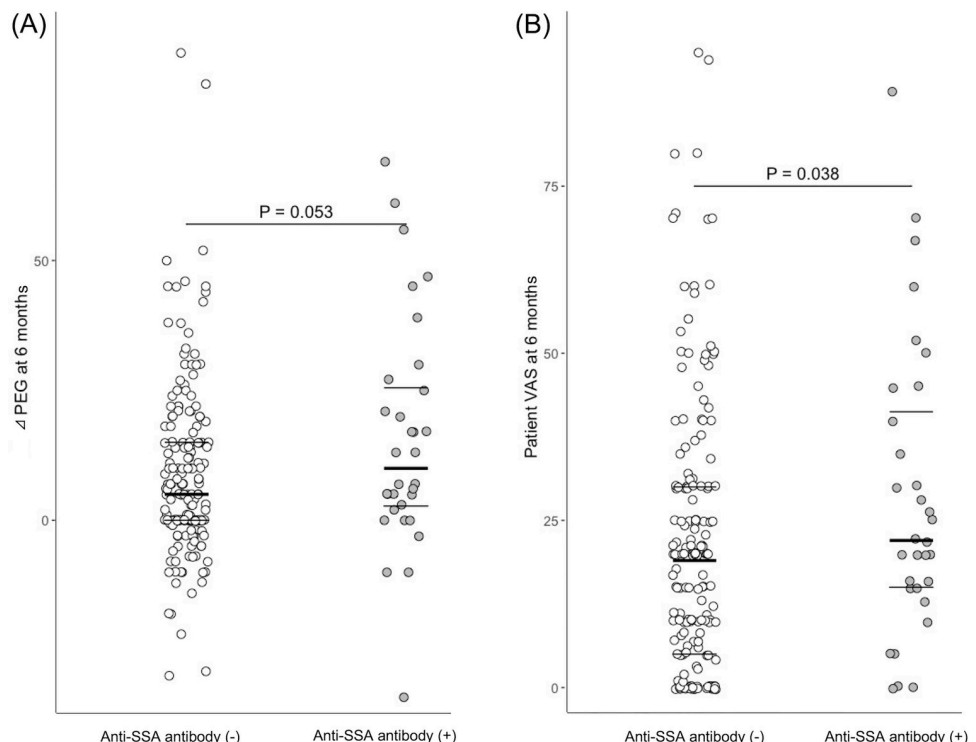

**Fig 1.** ΔPEG (A) and patient VAS score (B) after 6 months of MTX administration in the anti-SSA antibody-positive and -negative groups. Horizontal lines represent the median, 1st quartile, and 3rd quartile. ΔPEG, numeric difference between patient VAS score and physician VAS score; VAS, visual analogue scale.

which were significantly different between the groups after comparison of background factors. The presence of anti-SSA antibodies was still a considerable poor prognostic factor for achieving LDA based on the DAS28-CRP at 6 months (odds ratio: 0.419, 95% confidence interval: 0.182–0.961, $P = 0.040$) (Table 4).

To explore other potential risk factors, we compared baseline characteristics between the two groups that achieved LDA based on the DAS28-CRP at 6 months and those that did not (S1 Table). The results show that patients who did not achieve LDA had significantly higher baseline disease activity, more positivity rate of anti-SSA antibodies, and shorter disease duration. Based on these results, we performed logistic regression analysis including anti-SSA antibody positivity, IgM-RF positivity, anti-CCP antibody positivity, disease duration, and baseline DAS28-CRP activity. As a result, the presence of anti-SSA antibodies was still a considerable poor prognostic factor for achieving LDA based on the DAS28-CRP at 6 months (odds ratio: 0.406, 95% confidence interval: 0.174–0.949, P = 0.037) (Table 5).

## Discussion

Our study demonstrated that patients with anti-SSA antibody-positive RA are less responsive to initial MTX therapy. The anti-SSA antibody-positive group had higher patient VAS scores and higher ΔPEG values at 6 months, whereas the number of swollen and painful joints at 6 months was not significantly different between the groups. In the anti-SSA antibody-positive group, the rate of NSAID use was high at 6 months compared with that at baseline. These findings suggest that anti-SSA antibody-positive patients have more residual pain unrelated to joint tenderness.

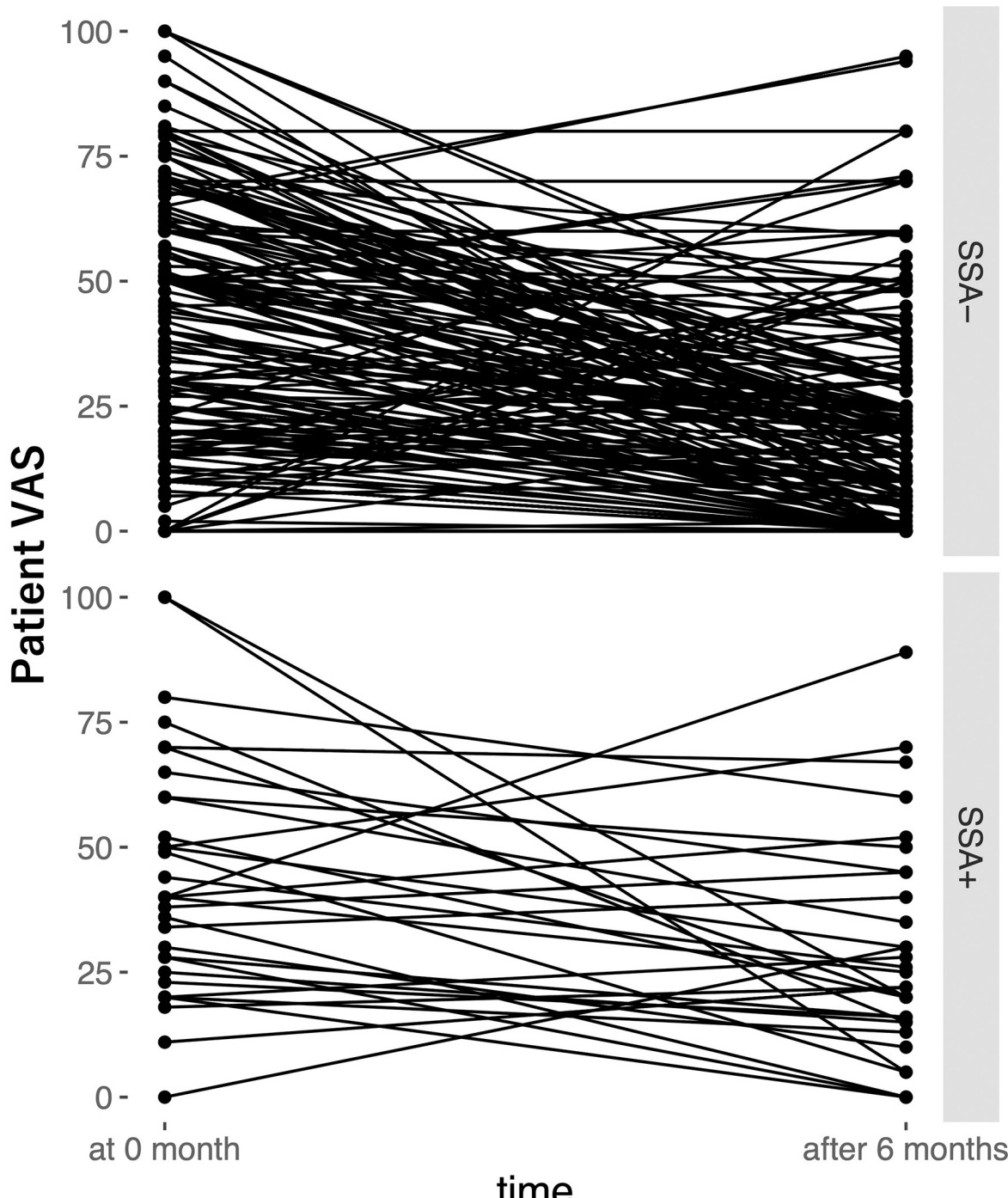

**Fig 2. Changing patient VAS score before and 6 months after MTX administration for each patient.**

The T2T approach is the cornerstone of treatment in RA, and MTX is its anchor drug. Although several poor prognostic factors for achieving T2T have been reported (e.g., anti-CCP antibodies, RF, disease duration, smoking, early bone erosion, and basal disease activity), whether anti-SSA antibodies are a poor prognostic factor has not been fully investigated. The results of our study suggest that the presence of anti-SSA antibodies may be a prognostic factor preventing the achievement of T2T.

**Table 3. Logistic regression analysis for the rate of achieving low disease activity based on the DAS28-CRP.**

| Risk factor | Odds ratio | 95% CI | P |
|---|---|---|---|
| Age at disease onset | 0.993 | 0.968–1.018 | 0.586 |
| Sex (woman) | 0.643 | 0.300–1.384 | 0.258 |
| IgM-RF positivity | 1.962 | 0.853–4.511 | 0.112 |
| Anti-CCP antibody positivity | 0.552 | 0.225–1.351 | 0.192 |
| Anti-SSA antibody positivity | 0.431 | 0.190–0.978 | 0.044 |

DAS28, 28-joint Disease Activity Score; CI, confidence interval; IgM-RF, IgM rheumatoid factor; anti-CCP, anti-cyclic citrullinated protein.

Two previous studies have reported the poor response to treatment with biologics in anti-SSA antibody-positive RA patients [10, 11]. In contrast, the cross-sectional study by Schneeberger et al. showed no significant difference in the treatment response between 14 patients with anti-SSA antibody-positive RA and 92 patients with anti-SSA antibody-negative RA. These studies included patients with varying treatment durations; therefore, the extent of the effect of treatment history in both groups cannot be determined [12]. In addition, these studies did not sufficiently analyze the influence of the potential risk factors.

In daily practice, the timing of starting bDMARDs after starting MTX therapy is varied because of the individual response to treatment. furthermore, some previous studies have suggested that bDMARDs may be less responsive in anti-SSA antibody-positive patients as mentioned above [10, 11]. We considered including such patients in our study would increase the difference between the two groups. Hence, we excluded patients who started bDMARDs within 6 months of the starting of the treatment, considering the variability in timing of treatment and the likelihood of differences. However, our results showed a significantly lower LDA rate achieving on the DAS28-CRP in the anti-SSA antibody-positive group. The median disease duration in our cohort was 5 months, and the low rate of csDMARD use meant that this population was less likely to be affected by previous treatment(s).

Some studies have shown that it may be difficult to treat patients with RA complicated by SS [23–25]. Genetics, epigenetics, Epstein-Barr virus infections, and effects on the adaptive immune system have been suggested as possible mechanisms. However, there is no strong evidence to support this and no clear indication of whether SS itself or anti-SSA antibodies affect disease activity. It has also been reported that bone destruction is more advanced in RA patients with SS. The effects on bone destruction could not be examined in this study; thus, further studies are needed to determine whether anti-SSA antibodies are more related to joint pain or bone destruction [24].

**Table 4. Logistic regression analysis for the rate of achieving low disease activity according to the DAS28-CRP, including the methotrexate dose.**

| Risk factor | Odds ratio | 95% CI | P |
|---|---|---|---|
| Methotrexate dose at 6 months | 0.968 | 0.877–1.070 | 0.533 |
| Sex (woman) | 0.656 | 0.307–1.404 | 0.277 |
| IgM-RF positivity | 1.923 | 0.840–4.403 | 0.121 |
| Anti-CCP antibody positivity | 0.607 | 0.252–1.459 | 0.192 |
| Anti-SSA antibody positivity | 0.419 | 0.182–0.961 | 0.040 |

DAS28, 28-joint Disease Activity Score; CI, confidence interval; IgM-RF, IgM rheumatoid factor; anti-CCP, anti-cyclic citrullinated protein.

**Table 5. Logistic regression analysis for rate of achieving low disease activity according to the DAS28-CRP, including baseline DAS28-CRP and disease duration.**

| Risk factor | Odds ratio | 95% CI | P |
|---|---|---|---|
| Baseline DAS28-CRP | 0.596 | 0.448–0.792 | < 0.001 |
| Disease duration | 0.998 | 0.993–1.004 | 0.998 |
| IgM-RF positivity | 2.326 | 0.969–5.580 | 0.058 |
| Anti-CCP antibody positivity | 0.384 | 0.150–0.983 | 0.046 |
| Anti-SSA antibody positivity | 0.406 | 0.174–0.949 | 0.037 |

Our results suggest that anti-SSA antibody-positive patients with RA have poor pain improvement after the initiation of MTX, affecting their disease activity. A large multicentric cohort study in Norway reported that discordance between patients' and evaluators' global assessment reduces the likelihood of clinical remission according to DAS28, SDAI, CDAI, and ACR/EULAR Boolean criteria in patients with RA [16]. Although these studies suggest that fibromyalgic RA may affect disease evaluations, no clear mechanism for this discordance has been described. Our results indicate that anti-SSA antibodies may contribute to this discordance. Unlike CDAI and SDAI, DAS28 does not include a Physician VAS [18]. This leads us to believe that variations in Patient VAS may have a relatively large impact on the overall index in DAS28. In addition, a higher weighting factor is assigned to tender joints compared to swollen joints in DAS28. Therefore, we speculate that the current results show a statistically significant difference only in the LDA achieving rate based on the DAS28-CRP.

Our results showed that there was a significant difference in MTX dosage between the two groups. However, the difference was about 1.0 mg on average. Considering the clinical efficacy, we believe it is unlikely that this result led to a treatment response in the anti-SSA antibody positive group. In fact, the logistic regression analysis accounting for MTX dosage also retained statistical significance for anti-SSA antibody as an independent poor prognostic factor for achieving LDA based on the DAS28-CRP (Table 4).

There are some strengths to our study. First, the multi-institutional approach reduced potential bias caused by the characteristics of the research institution and the subjectivity of the researcher. Second, missing data were imputed to improve the statistical power of our study. Third, the majority of the included patients were in the early phase of RA, which reduced the possible effects of prior drug administration and prolonged disease duration. Finally, we conducted multivariable analyses considering age, sex, RF, and anti-CCP antibody status, which are risk factors influencing the response to RA treatment.

We also acknowledge several limitations of our study. First, given the retrospective observational nature of the study, selection and memory biases cannot be completely excluded. As a potential selection bias, patients whose anti-SSA antibody titer was not available, patients who discontinued MTX, and patients who started biologics were excluded from the study. However, the rate of anti-SSA antibody positivity in this study was comparable to that in previous studies (3%–15%), and the fact that the physicians arranged for anti-SSA antibody measurements suggests the study population was more prone to SS with minor variation in confounders not assessed in this study. Similarly, it is unlikely that the treatment had changed depending on the presence or absence of anti-SSA antibodies. In addition, we believe that limiting the treatment options made it possible to standardize the treatment within our cohort. Second, we were unable to collect information regarding indicators of structural damage, such as the total sharp score, and indicators of the patients' quality of life. Third, Long-term effects of anti-SSA antibodies on the treatment of RA are not clear, since the observation period of this study was 6 months. Finally, we did not determine the prevalence of SS in our study cohort

because many patients did not receive sufficient testing to diagnose SS. Therefore, we could not investigate in this study whether SS diagnosis or anti-SSA antibody positivity has more influence on disease activity. However, many patients with SS are positive for anti-SSA antibodies, and we believe that it is highly likely that anti-SSA antibodies themselves affect the disease activity.

## Conclusions

The presence of anti-SSA antibodies could be a risk factor influencing the response to conventional RA treatment through residual pain. Further studies are warranted to determine whether it is beneficial to effectively treat patients with anti-SSA antibody-positive RA, including the effects on joint destruction and quality of life.

## Supporting information

**S1 Table. Patient characteristics at baseline between the patients who achieve LDA based on the DAS28-CRP and those that did not.**
(DOCX)

## Acknowledgments

The authors wish to thank Dr Hisashi Noma, Department of Data Science, The Institute of Statistical Mathematics, for his support in confirming the validity and robustness of the data analysis. The authors acknowledge the staff and the participants in the first clinical research workshop by the Japan College of Rheumatology for their helpful comments regarding the study design.

## Author Contributions

**Conceptualization:** Hiroya Tamai, Ritsuko Yokochi, Toshiki Kido, Yuriko Yagyu, Ryo Yanai, Ken-Ei Sada.

**Data curation:** Daisuke Waki, Hiroya Tamai, Ritsuko Yokochi, Toshiki Kido.

**Formal analysis:** Daisuke Waki.

**Investigation:** Hiroya Tamai, Ritsuko Yokochi, Toshiki Kido.

**Validation:** Daisuke Waki, Hiroya Tamai, Ritsuko Yokochi, Ryo Yanai, Ken-Ei Sada.

**Writing – original draft:** Daisuke Waki, Yuriko Yagyu.

**Writing – review & editing:** Daisuke Waki, Hiroya Tamai, Ritsuko Yokochi, Toshiki Kido, Yuriko Yagyu, Ryo Yanai, Ken-Ei Sada.

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
