## [Decision Letter · Decision Letter 0]

18 May 2022

PONE-D-22-11472Effects of anti-SSA antibodies on the response to methotrexate in rheumatoid arthritis: a retrospective multicenter observational studyPLOS ONE

Dear Dr. waki,

Thank you for submitting your manuscript to PLOS ONE. After careful consideration, we feel that it has merit but does not fully meet PLOS ONE’s publication criteria as it currently stands. Therefore, we invite you to submit a revised version of the manuscript that addresses the points raised during the review process.

Our reviewers found some interests in this study, but also pointed out a number of issues that require amendment or improvement, including entirely new statistical analyses. I ask the authors to fully respond to all comments made by reviewers in the revised version.  

We look forward to receiving your revised manuscript.

Kind regards,

Masataka Kuwana, MD, PhD

Academic Editor

PLOS ONE

Journal Requirements:

2. We note that your study used an opt-out consent mechanism. Please state whether the ethics committee approved this consent mechanism.

Reviewers' comments:

Reviewer's Responses to Questions

**Comments to the Author**

1. Is the manuscript technically sound, and do the data support the conclusions?

Reviewer #1: Partly

Reviewer #2: Yes

2. Has the statistical analysis been performed appropriately and rigorously? 

Reviewer #1: Yes

Reviewer #2: Yes

3. Have the authors made all data underlying the findings in their manuscript fully available?

Reviewer #1: Yes

Reviewer #2: Yes

4. Is the manuscript presented in an intelligible fashion and written in standard English?

Reviewer #1: Yes

Reviewer #2: Yes

5. Review Comments to the Author

Reviewer #1: This manuscript is a multicenter retrospective study of early-onset MTX-naive patients.

The response to MTX treatment in SS-A antibody-positive RA patients was compared with that in SS-A antibody-negative patients.

There were significant differences in patients' VAS, DAS28CRP LDA achievement rate, and history of NSAIDs use, suggesting that SS-A antibody positivity may be a predictor of inadequate treatment response.

This conclusion in not fully supported by the data.

The study has many limitations, and it is difficult to conclude from the results of this manuscript that it is a risk factor for inadequate treatment response.

The authors' inadequate treatment response excludes patients who progressed to bDMARD treatment, which is a major limitation in the interpretation of the study results.

The observation period is only 6 months and cannot be referred to as a long-term risk factor.

Major Revisions

Describe the validity of defining inadequate treatment as no significant difference in SDAI and CDAI, but only in patient VAS, NSAIDs use, and DAS28CRP LDA.

It should be demonstrated that the lower average dose of MTX in SS-A positive RA does not affect the interpretation of the results.

For each patient, the change in patient VAS should be plotted before and 6 months after MTX administration.

If the focus is on patient VAS, include patient VAS as an explanatory variable in the logistic regression analysis

Table 1 should present the incidence of extra-articular complications.

Reviewer #2: In this manuscript, the authors reported the multicenter retrospective cohort study in which a total of 210 consecutive patients with rheumatoid arthritis (RA) who newly initiated methotrexate (MTX) were recruited. This study involved 32 and 178 anti-SSA antibody-positive and -negative patients, respectively. The rate of achieving low disease activity (LDA) according to the 28-joint Disease Activity Score based on C-reactive protein (DAS28-CRP) at 6 months was significantly lower in the anti-SSA antibody-positive group than in the anti-SSA antibody-negative group. The multivariable logistic regression analysis showed that anti-SSA antibody positivity was significantly associated with failure to achieve LDA according to the DAS28-CRP at 6 months, even after adjusting for the potential poor prognostic factors. The authors concluded that the presence of anti-SSA antibodies might be a predictive factor for insufficient responses to treat-to-target (T2T) strategy in RA, and residual pain might contribute to the reduced clinical response to MTX in anti-SSA antibody-positive patients with RA. Although these findings might be clinically important and meaningful, the authors should address following points.

1) In the anti-SSA antibody-positive group, the dose of MTX at 6 months was significantly lower than in the anti-SSA antibody-negative group. The reason why MTX doses were lower in the anti-SSA antibody-positive group should be discussed.

2) Before multivariable logistic regression analysis, the comparison of variables between patients who achieved LDA according to the DAS28-CRP at 6 months and patients who did not achieve might be informative.

3) The disease duration and disease activity at baseline could be the potential poor prognostic factors. Thus, these factors should be also examined by multivariable logistic regression analysis.

6. PLOS authors have the option to publish the peer review history of their article (what does this mean?). If published, this will include your full peer review and any attached files.

Reviewer #1: No

Reviewer #2: No

---

## [Author Response · Author response to Decision Letter 0]

13 Jun 2022

Reviewer #1’s comments

There were significant differences in patients' VAS, DAS28CRP LDA achievement rate, and history of NSAIDs use, suggesting that SS-A antibody positivity may be a predictor of inadequate treatment response. This conclusion is not fully supported by the data. The study has many limitations, and it is difficult to conclude from the results of this manuscript that it is a risk factor for inadequate treatment response.

As Reviewer #1 points out, we do not believe that our results can be used to conclude that anti-SSA antibodies are a poor prognostic factor for treatment of RA, but our results indicate that anti-SSA antibodies may be a predictor of poor response to conventional MTX therapy, independent of other established poor prognostic factors.

The authors' inadequate treatment response excludes patients who progressed to bDMARD treatment, which is a major limitation in the interpretation of the study results. 

In daily practice, the timing of starting bDMARDs after starting MTX therapy is varied because of the individual response to treatment. In addition, some previous studies have suggested that bDMARDs may be less responsive in anti-SSA antibody-positive patients. We considered including such patients in our study would increase the difference between the two groups. Therefore, we excluded patients who started bDMARDs within 6 months of the start of treatment, considering the variability in timing of treatment and the likelihood of differences. However, the results showed a significantly lower DAS28-LDA rate in the anti-SSA antibody-positive group. We added the sentence about this point in the Discussion section.

Changes: The Discussion section of the manuscript now contains the following sentence: 

“In daily practice, the timing of starting bDMARDs after starting MTX therapy is varied because of the individual response to treatment. furthermore, some previous studies have suggested that bDMARDs may be less responsive in anti-SSA antibody-positive patients as mentioned above [10,11]. We considered including such patients in our study would increase the difference between the two groups. Hence, we excluded patients who started bDMARDs within 6 months of the starting of the treatment, considering the variability in timing of treatment and the likelihood of differences. However, our results showed a significantly lower LDA rate achieving based on the DAS28-CRP in the anti-SSA antibody-positive group.” (Page 19, Line 241-248)

The observation period is only 6 months and cannot be referred to as a long-term risk factor.

We agree with you that our results cannot be referred as a long-term effect for treatment response of RA. We added the sentence about this limitation in the Discussion section.

Changes: The Discussion section of the manuscript now contains the following sentence: 

“Third, Long-term effects of anti-SSA antibodies on the treatment of RA are not clear, since the observation period of this study was 6 months.” (Page 22, Line 291-293)

Major Revisions Describe the validity of defining inadequate treatment as no significant difference in SDAI and CDAI, but only in patient VAS, NSAIDs use, and DAS28CRP LDA.

Unlike CDAI and SDAI, DAS28 does not include a Physician VAS. This leads us to believe that variations in Patient VAS may have a relatively large impact on the overall index in DAS28. In addition, a higher weighting factor is assigned to tender joints compared to swollen joints in DAS28 .Therefore, we speculate that the current results show a statistically significant difference only in the LDA achieving rate for DAS28. We added the sentence about this hypothesis in the Discussion section.

As mentioned in the first paragraph of Discussion section, we believe that the difference in NSAIDs use at 6 months was due to the residual pain remaining more in the SSA-positive group.

Changes: The Discussion section of the manuscript now contains the following sentence: 

“Unlike CDAI and SDAI, DAS28 does not include a Physician VAS [18]. This leads us to believe that variations in Patient VAS may have a relatively large impact on the overall index in DAS28. In addition, a higher weighting factor is assigned to tender joints compared to swollen joints in DAS28. Therefore, we speculate that the current results show a statistically significant difference only in the LDA achieving rate based on the DAS28-CRP.” (Page 20, Line 263-267)

It should be demonstrated that the lower average dose of MTX in SS-A positive RA does not affect the interpretation of the results.

As Reviewer #1 pointed out, there was a significant difference in MTX dosage between the two groups in this analysis, but the difference was about 1.0 mg on average. Taking into account the clinical efficacy, we believe it is unlikely that this result led to a treatment response in the SSA-positive group.

In fact, the logistic regression analysis (Table 4) accounting for MTX dosage also retained statistical significance for anti-SSA antibody positivity as an independent poor prognostic factor for achieving LDA in DAS28-CRP. we have added our consideration of this difference in the Discussion section.

Changes: The Discussion section of the manuscript now contains the following sentence: 

“Our results showed that there was a significant difference in MTX dosage between the two groups. However, the difference was about 1.0 mg on average. Considering the clinical efficacy, we believe it is unlikely that this result led to a treatment response in the anti-SSA positive group. In fact, the logistic regression analysis accounting for MTX dosage also retained statistical significance for anti-SSA antibody as an independent poor prognostic factor for achieving LDA based on the DAS28-CRP (Table 4).” (Page 21, Line 268-273)

For each patient, the change in patient VAS should be plotted before and 6 months after MTX administration.

According to Reviewer #1’s advice, we have added the new figure as Fig 2. (Page 12, Line 166-167)

If the focus is on patient VAS, include patient VAS as an explanatory variable in the logistic regression analysis

Our hypothesis is that patient VAS should be an intermediate factor regarding anti-SSA antibodies as the prognostic factor and achieving LDA based on the DAS28-CRP as the outcome, therefore, we believe that a multivariate analysis adding this as a variable is not appropriate from a statistical perspective.

Table 1 should present the incidence of extra-articular complications.

Patients with extra-articular complications were excluded from this study, and we have added a note about this in the Materials and methods section. Thank you for clarifying this point.

Changes: The Materials and methods section of the manuscript now contains the following sentence: 

“Patients with extra-articular complications were also excluded from this study.” (Page 5, Line 85)

 

Reviewere #2’s comments

1) In the anti-SSA antibody-positive group, the dose of MTX at 6 months was significantly lower than in the anti-SSA antibody-negative group. The reason why MTX doses were lower in the anti-SSA antibody-positive group should be discussed.

As we discussed above in our response to Reviewer #1, although there was a significant difference in MTX dosage between the two groups in this analysis, the difference was about 1.0 mg on average. Taking into account the clinical efficacy, we believe it is unlikely that this result led to a treatment response in the SSA-positive group. However, it is true that there was a significant difference, so we have added our consideration of this difference in the Discussion section.

Changes: The Discussion section of the manuscript now contains the following sentence: 

“Our results showed that there was a significant difference in MTX dosage between the two groups. However, the difference was about 1.0 mg on average. Considering the clinical efficacy, we believe it is unlikely that this result led to a treatment response in the SSA-positive group. In fact, the logistic regression analysis accounting for MTX dosage also retained statistical significance for anti SSA antibody as an independent poor prognostic factor for achieving LDA based on the DAS28-CRP (Table 4).” (Page 21, Line 268-273)

Before multivariable logistic regression analysis, the comparison of variables between patients who achieved LDA according to the DAS28-CRP at 6 months and patients who did not achieve might be informative.

In response to Reviewer #2, we performed an additional analysis comparing baseline characteristics by dividing patients into two groups: patients who achieved LDA based on the DAS28-CRP at 6 months and those who did not. The results are presented in S1 Table (Page 27, Line 396-398). The results show that patients who did not achieve LDA had higher baseline disease activity, more positivity rate of anti-SSA antibodies, and shorter disease duration.

Changes: The Supporting information section of the manuscript now contains the following sentence: 

“Supporting information

S1 Table. Patient characteristics at baseline between the patients who Achieve LDA based on the DAS28-CRP and those that did not.” (Page 27, Line 396-398).

3) The disease duration and disease activity at baseline could be the potential poor prognostic factors. Thus, these factors should be also examined by multivariable logistic regression analysis.

As pointed out by Reviewer #2, the results of S1 Table suggest that disease activity at baseline and disease duration may be potential poor prognostic factors for achieving LDA based on the DAS28-CRP. Considering the sample size in this study, we could not perform a multivariate analysis including all potential prognostic factors. Hence, we performed logistic regression analysis for achieving LDA based on the DAS28-CRP including anti-SSA antibody, IgM-RF, anti-CCP antibody, disease duration, and baseline DAS28-CRP activity after multiple imputations. As a result, the presence of anti-SSA antibodies was still a considerable poor prognostic factor for achieving LDA based on the DAS28-CRP at 6 months (odds ratio: 0.406, 95% confidence interval: 0.174–0.949, P = 0.037) (Table 5).

Changes: The Material and Methods section of the manuscript now contains the following sentence: 

“To explore other potential prognostic factors, we compared baseline characteristics between the two groups that achieved LDA based on DAS28-CRP at 6 months and those that did not. We performed another sensitivity analysis considering those potential prognostic factors.” (Page 7, Line 126-129).

Changes: The Results section of the manuscript now contains the following sentence: 

“To explore other potential prognostic factors, we compared baseline characteristics between the two groups that achieved LDA based on the DAS28-CRP at 6 months and those that did not (S1 Table). The results show that patients who did not achieve LDA had significantly higher baseline disease activity, more positivity rate of anti-SSA antibodies, and shorter disease duration. Based on these results, we performed logistic regression analysis including anti-SSA antibody positivity, IgM-RF positivity, anti-CCP antibody positivity, disease duration, and baseline DAS28-CRP activity. As a result, the presence of anti-SSA antibodies was still a considerable poor prognostic factor for achieving LDA based on the DAS28-CRP at 6 months (odds ratio: 0.406, 95% confidence interval: 0.174–0.949, P = 0.037) (Table 5).” (Page 17, Line 208-216).

Table 5. Logistic regression analysis for the rate of achieving low disease activity according to the DAS28-CRP, including baseline DAS28-CRP and disease duration.

 Risk factor Odds ratio 95% CI P

Baseline DAS28-CRP 0.596 0.448–0.792 < 0.001

Disease duration 0.998 0.993–1.004 0.998

IgM-RF positivity 2.326 0.969–5.580 0.058

Anti-CCP antibody positivity 0.384 0.150–0.983 0.046

Anti-SSA antibody positivity 0.406 0.174–0.949 0.037

---

## [Decision Letter · Decision Letter 1]

11 Jul 2022

Effects of anti-SSA antibodies on the response to methotrexate in rheumatoid arthritis: a retrospective multicenter observational study

PONE-D-22-11472R1

Dear Dr. waki,

We’re pleased to inform you that your manuscript has been judged scientifically suitable for publication and will be formally accepted for publication once it meets all outstanding technical requirements.

Kind regards,

Masataka Kuwana, MD, PhD

Academic Editor

PLOS ONE

Additional Editor Comments (optional):

All comments are adequately answered in the revised version.

Reviewers' comments:

Reviewer's Responses to Questions

**Comments to the Author**

1. If the authors have adequately addressed your comments raised in a previous round of review and you feel that this manuscript is now acceptable for publication, you may indicate that here to bypass the “Comments to the Author” section, enter your conflict of interest statement in the “Confidential to Editor” section, and submit your "Accept" recommendation.

Reviewer #2: All comments have been addressed

2. Is the manuscript technically sound, and do the data support the conclusions?

Reviewer #2: Yes

3. Has the statistical analysis been performed appropriately and rigorously? 

Reviewer #2: Yes

4. Have the authors made all data underlying the findings in their manuscript fully available?

Reviewer #2: Yes

5. Is the manuscript presented in an intelligible fashion and written in standard English?

Reviewer #2: Yes

6. Review Comments to the Author

Reviewer #2: Authors have successfully revised the manuscript according to reviewers’ comments. I have no further comments.

7. PLOS authors have the option to publish the peer review history of their article (what does this mean?). If published, this will include your full peer review and any attached files.

Reviewer #2: No

---

## [Editor Report · Acceptance letter]

14 Jul 2022

PONE-D-22-11472R1 

Effects of anti-SSA antibodies on the response to methotrexate in rheumatoid arthritis: a retrospective multicenter observational study 

Dear Dr. waki:

I'm pleased to inform you that your manuscript has been deemed suitable for publication in PLOS ONE. Congratulations! Your manuscript is now with our production department. 

Kind regards, 

on behalf of

Prof. Masataka Kuwana 

Academic Editor

PLOS ONE